# The Genetic Architecture of Milling Quality in Spring Oat Lines of the Collaborative Oat Research Enterprise

**DOI:** 10.3390/foods10102479

**Published:** 2021-10-16

**Authors:** Kathy Esvelt Klos, Belayneh A. Yimer, Catherine J. Howarth, Michael S. McMullen, Mark E. Sorrells, Nicholas A. Tinker, Weikai Yan, Aaron D. Beattie

**Affiliations:** 1Small Grains and Potato Germplasm Research Unit, United States Department of Agriculture-Agricultural Research Service (USDA-ARS), 1691 South 2700 West, Aberdeen, ID 83210, USA; 2Department of Plant, Soil, and Entomological Sciences, University of Idaho Research and Extension, Idaho Falls, ID 83210, USA; byimer@uidaho.edu; 3Institute of Biological, Environmental and Rural Sciences, Aberystwyth University, Gogerddan, Aberystwyth SY23 3EE, UK; cnh@aber.ac.uk; 4Department of Plant Sciences, North Dakota State University, P.O. Box 6050, Fargo, ND 58108, USA; Michael.Mcmullen@ndsu.edu; 5Plant Breeding and Genetics, Cornell University, 240 Emerson Hall, Ithaca, NY 14853, USA; mes12@cornell.edu; 6Agriculture and AgriFoods Canada (AAFC), Ottawa Research and Development Centre, 960 Carling Ace., Central Experiment Farm, Ottawa, ON K1A 0C6, Canada; nick.tinker@canada.ca (N.A.T.); Weikai.yan@canada.ca (W.Y.); 7Crop Development Centre, University of Saskatchewan, 51 Campus Drive, Saskatoon, SK S7N 5A8, Canada; aaron.beattie@usask.edu

**Keywords:** oat, milling, GWAS, QTL

## Abstract

Most oat grains destined for human consumption must possess the ability to pass through an industrial de-hulling process with minimal breakage and waste. Uniform grain size and a high groat to hull ratio are desirable traits related to milling performance. The purpose of this study was to characterize the genetic architecture of traits related to milling quality by identifying quantitative trait loci (QTL) contributing to variation among a diverse collection of elite and foundational spring oat lines important to North American oat breeding programs. A total of 501 lines from the Collaborative Oat Research Enterprise (CORE) panel were evaluated for genome-wide association with 6 key milling traits. Traits were evaluated in 13 location years. Associations for 36,315 markers were evaluated for trait means across and within location years, as well as trait variance across location years, which was used to assess trait stability. Fifty-seven QTL influencing one or more of the milling quality related traits were identified, with fourteen QTL mapped influencing mean and variance across location years. The most prominent QTL was Qkernel.CORE.4D on chromosome 4D at approximately 212 cM, which influenced the mean levels of all traits. QTL were identified that influenced trait variance but not mean, trait mean only and both.

## 1. Introduction

Oat (*Avena sativa* L.) grain production in North America is primarily targeted towards use as human food. The value of oat varies throughout the oat value chain. To a consumer, value may include such considerations as convenience, taste, and the acknowledged cardiovascular health benefits (reviewed by [1]). Value to growers and industrial suppliers of oat is more easily couched in monetary terms. According to the Foreign Agricultural Service monthly reports on World Agricultural Production (www.fas.usda.gov/commodites/oats; 10 January 2018), North American oat growers have produced an average of 4.25 million metric tons (292.87 million bushels) of oat grain per year over the past decade. The price per bushel over the same time ranged from USD 1.92 to USD 4.72, suggesting an annual gross value of approximately USD 878 million. Of course, the profit to growers is highly dependent on the cost of inputs, which are variable. Cultivar characteristics that maximize value from a grower’s perspective include grain yield, straw strength and biotic/abiotic stress resistance. Industrial suppliers of oat for human food consider the values of the product, co-products and by-products after deducting the cost of raw oat and the cost of processing. Value in this context is highly influenced by the capacity and efficiency of the milling equipment and processes [2]. Grain coming into the mill for processing as human food is separated and graded according to size, shape and density, resulting in the rejection of thin, light and poorly developed kernels [3,4]. Assuming a milling volume of 20 million bushels and a cost of USD 3.00 per US bushel, then the value of improving the proportion of groat to hull ratio (the mill yield), is approximately USD 250,000 per percent improvement [2]. A desirable cultivar will produce seed that is plump, uniform in width and length to minimize the proportion of rejected kernels at cleaning and will have a high groat to hull ratio.

The quality parameters used by oat millers to evaluate cultivars for desirability include the groat to hull ratio, ease of dehulling, uniformity of groat size, uniformity of mature kernels from the top to bottom of the panicle, overall groat size in the medium to large range, few trichomes, trichomes that are easily separated and a groat lacking a deep crease that is prone to discoloration [2]. In an experimental setting, small-scale technologies are used to approximate conditions in a milling plant. For example, groat content (GC) is the groat percentage by weight after mechanical dehulling and is reflective of the groat to hull ratio. Thousand kernel weight (TKW) reflects kernel density and depends on kernel size and the ratio of groat to hull. The percent plump kernels (Plumps) and percent thin kernels (Thins) describe the proportion of kernels retained or passing through sieves of specific sizes and reflect overall kernel size and uniformity. Thus, milling quality characteristics are not independent of one-another, but rather are correlated due to shared environmental and genetic influences. Previous studies have estimated phenotypic correlation between GC and TKW is -0.50 to -0.58, between GC and test weight (TWT) is 0.59 to 0.68, between Plumps and TKW is 0.35 to 0.66 and between TKW and TWT is 0.35 to 0.64 [5,6,7,8]. The heritability of GC has been estimated at 0.88 to 1.00, Plumps at 0.92, Thins at 0.97, TKW at 0.88 to 0.98 and TWT at 0.60 to 0.97 [6,9,10,11,12].

Documented environmental influences on oat milling quality include the negative influence of crown rust infection on the percent of broken groats, GC, Thins and TWT [13], as well as the positive influence of environments that promote high grain yield on GC, TWT, TKW and Plumps [5,14]. Peterson et al. [11] observed significant genotype by year and phenotype by environment by year interaction effects influencing kernel weight and groat percentage in a set of 33 lines grown in 9 location years in Idaho, although the main genotype effects were much greater. Doehlert and McMullen [15] theorized that common factors underlying the response to environmental stress may influence groat size and percentage, reflecting processes active during grain filling.

Previous linkage and association mapping efforts have identified quantitative trait loci (QTL) influencing milling quality traits. Linkage mapping in the Kanota × Ogle and Kanota × Marion populations identified kernel morphology, GC and TWT QTL [16,17]. Linkage mapping in the Terra × Marion population identified 10 QTL on 9 linkage groups influencing percent plumps, percent thins, GC, TKW and TWT, singly or in combination [10]. This set included a QTL linked to markers now placed to chromosome 4D [18] that influenced all five of those traits. Using two backcross populations, Herrmann et al. [7] mapped three GC, four TKW and three TWT QTL, although the QTL on chromosome 7A at approximately 30 to 65 cM may have represented a single pleotropic effect [18,19]. Zimmer et al. [20] identified three QTL influencing kernel length and a QTL influencing both width and length in an association mapping panel of oat adapted to subtropical environments.

The purpose of this study was to identify regions of the oat genome contributing to variation in important milling traits indicative of kernel density, kernel size and the ability of groats to withstand mechanical dehulling. These qualities were assessed in a representative sample of elite oat cultivars and other lines foundational to modern oat breeding programs of North America across 13 location years representing major regions producing milling quality oats [21].

## 2. Materials and Methods

### 2.1. Plant Materials and Molecular Markers

Sampling strategies, genotyping methods and quality control procedures applied to the Collaborative Oat Research Enterprise (CORE) panel have been described elsewhere and are summarized below [21]. The CORE association panel consists of 635 single panicle-derived lines representative of elite germplasm deemed important by 16 active oat breeding programs in Australia, Canada, the U.K. and the U.S.A. However, lines nominated to the association mapping panel by oat breeders working in the southern U.S. were not evaluated for milling quality at all location years. To achieve a more balanced dataset across environments, only those spring-planted lines nominated to the CORE as part of the world diversity panel and those nominated by spring oat breeders were included in this study (501 lines). All CORE lines were characterized using the oat iSelect 6K-beadchip array for 4328 polymorphic SNPs with <5% heterozygosity, missing calls <5% and a minor allele frequency of >1%. Genotyping-by-sequencing (GBS) SNPs were generated as described in Esvelt Klos et al. [21] and tag-level haplotype loci (also referred to below as SNPs) inferred as described by Bekele et al. [19]. GBS SNPs were removed from the analysis dataset if missing data was observed in >40% of the lines, the minor allele frequency was <1% and heterozygosity was >5%. Genotypes of some SNPs were identical across all lines, and only one SNP was retained for analysis with preference given to those with map locations. There was a total of 36,315 SNPs retained for statistical analyses. Map locations of SNPs were based on the consensus map of Chaffin et al. [22], as expanded by Bekele et al. [19], and used the chromosome designations of the *Avena sativa*—OT3098 v2 genome (PepsiCo, https://wheat.pw.usda.gov/jb?data=/ggds/oat-ot3098v2-pepsico; 4 January 2021).

### 2.2. Phenotyping

Milling-quality-related trait data were taken on grain produced at Aberdeen, ID in 2010 and 2017 (Ad10 and Ad17); Aberystwyth, UK in 2010 (Ay10); Fargo, ND in 2010 and 2011 (Fa10 and Fa11); Ithaca 2011 (It11); Lacombe, AB in 2010 and 2011 (La10 and La11); Ottawa, ON in 2010 (Ot10); Saskatoon, SK in 2010 and 2011 (Sa10 and Sa11); and Tetonia, ID in 2010 and 2011 (Te10 and Te11; Table 1). The details pertaining to planting and harvesting can be found at the public T3/oat database (http://triticeaetoolbox.org/oat/; 1 September 2021) and summarized by Esvelt Klos et al. [21]. Additionally, the CORE lines evaluated at Aberdeen, ID in 2017 were planted the week of 15 May 2017 in unreplicated 1.8 m headrows and harvested the week of 4 September 2017.

The percent of broken groats (Br%; estimated visually) and groat content (GC, the percentage by weight of groats in relation to grain) were assessed on approximately 50g of grain after de-hulling for 60 s at 90 psi with a laboratory oat huller (Model LH 5095, Codema Inc., Eden Prairie, MN). Grain was evaluated for width by passing through slotted sieves (Seedburo Equipment Co., Chicago, IL, USA) and recording the percentage by weight retained on a 2.18 × 19.05-mm slotted sieve (Plumps) and passing through a 1.98 × 19.05-mm slotted sieve (Thins). Thousand kernel weight (TKW) was recorded in grams. Test Weight (TWT) was determined by passing grain through a Cox funnel into a 0.5-L cylindrical cup, leveling off excess grain from the top of the cup and weighing the resulting grain (g) retained within the cup before converting to kg/hL. The distributions of percent broken groats and percent thins were highly skewed (Appendix A), so these traits were log10-transformed which resulted in a more normal distribution.

### 2.3. Statistical Analyses

The arithmetic means across all available location years were estimated for each line in SAS v. 9.4 (SAS Institute, Gary, NC, USA). Trait stability across production environments was estimated as the variance (the average of the total squared dispersion between observations and the sample mean). Although the number and diversity of location years was large, the interpretation of this trait beyond these should be made with caution. The effect of genotype on milling quality and the effect of location year on milling quality were evaluated by one-way ANOVA in SAS v. 9.4 (SAS Institute, Gary, NC, USA), with equality of variances tested using the Levene’s test and the Welch’s statistic used where variances were unequal. Pairwise differences between location year means were evaluated using Fisher’s least significant difference. Pearson’s correlation was used to describe the relationships between traits with a *p*-value threshold of ≤0.01 used to establish significance.

Single-SNP association analyses were performed in TASSEL v5.0 [23] under the default settings. Missing genotype data were numerically imputed using data from the five nearest neighbors and computed by the Euclidean distance. Mixed linear models (MLM) incorporating a kinship matrix alone or with the first three principal components (PC) were compared with general linear models (GLM) incorporating the first 3 PCs for each experiment. Quantile-quantile (Q-Q) plots were used to compare statistical models for their ability to correct the tendency for Type I error inflation due to population structure and cryptic relatedness within the sample. Although results varied, MLM incorporating 3 PCs and a kinship matrix was selected as the final model for analyses of all variables because this provided the best adjustment for type 1 error inflation (not shown). Statistical significance was taken as *p* ≤ 1.41 × 10^−6^, based on the Bonferroni adjustment to maintain α = 0.05 in tests of multiple SNPs.

Where multiple SNPs are associated with phenotype, they may be capturing genotype information at the same QTL or they may represent independent effects. Incorporation of marker genotype into the MLM model as an additional covariate was expected to reduce evidence of association at other non-independent SNPs (i.e., at the SNPs that capture genotype information related to the same QTL). To evaluate SNPs for statistical non-independence, association models were run which incorporated one significant SNP as a covariate. Previously significant associations with *p*-values > 0.001 under the new model were taken as non-independent of the SNP used as covariate. Genotype-phenotype associations were investigated for non-independence in order of *p*-value, with non-independent associations not considered thereafter. The reported representative marker for each QTL were selected from among all statistically non-independent SNPs as the SNP with the lowest *p*-value for association with trait mean, trait variance or trait level within location year, as appropriate. Post-hoc evaluation to test specific hypotheses regarding how the number of QTL associated with specific milling traits affected the mean of those traits were evaluated in TASSEL using the MLM with 3 PCs as described above. The number of QTL, as indicated by classification as a homozygous rare allele carrier at the marker deemed most representative of the QTL, was characterized by classifying each line as carrying 0, 1 or >1 QTL and re-coding these categories as genotypes in the hapmap format. For significant associations, genotype or category trait means were determined in SAS v. 9.4.

## 3. Results

The six measures of milling quality evaluated in this study were Br%, GC, Plumps, Thins, TKW and TWT (Table 1). The r^2^ values (the explained variance) from general linear models fitting line (genotype) were 0.19, 0.47, 0.33, 0.35, 0.49 and 0.24 for Br%, GC, Plumps, Thins, TKW and TWT, respectively. The line effect was significant for all measures (*p* < 0.0001). When location year was fitted in a general linear model, this effect was significant for all measures (*p* < 0.0001). The trait phenotypic correlations between the mean and variance for each of the GC, Plumps, Thins and TWT were all negative (Table 2). Correlations between trait variances were not significant while correlations between trait means ranged from −0.93 (Plumps and logThins) to 0.76 (Plumps and TKW).

Statistically significant association (*p* ≤ 1.41 × 10^−6^) was observed between 1158 SNPs and at least 1 phenotype at 1 location year (Appendix A). The tests of statistical non-independence reduced this to 57 QTL influencing milling quality. The post-hoc examination of association in all experiments was used to divide QTL into three groups. Our primary interest was in identifying the map locations of QTL influencing milling quality across location years, so Table 3 lists the 14 QTL with statistically significant trait mean and/or trait variance association(s). These have been named based on the trait(s) for which statistically significant association was detected, the CORE association mapping panel (for ease of comparison with QTL from other studies) and their location on the oat consensus map. A second group of 17 QTL (Table 4) consisted of those influencing trait means and variances but which were represented by SNPs without locations on the consensus map or by SNPs mapped to more than one linkage group. Finally, we identified 26 QTL (Table 5) whose effect was on trait within 1 location year. The naming of these QTL incorporated information on location year and map location, if available

## 4. Discussion

The current study evaluated the genotype–phenotype association in breeder-nominated spring oat lines and identified 57 independent effects on milling quality trait mean, variance and/or mean within location year. These findings allow for a comparison of the mapped QTL to previously published milling quality QTL and their relationship with QTL for correlated traits. In addition, the genetic architecture of oat milling can be examined in terms of the relationships between trait level and trait stability, patterns of influence on phenotype and the additivity of main effects, all of which are important considerations for oat breeders to understand.

Of the QTL identified, 32 were unable to be placed to positions on the consensus map with certainty. In 12 cases, this was because none of the markers associated with the QTL were included in the consensus map. It is hoped that the future annotation of the oat genome will address this limitation and allow placement of these markers. The remaining unmapped QTL consisted of groups of non-independent markers that mapped to multiple linkage groups on the consensus map. In GWAS, this can happen when a relatively rare variant is responsible for changes in phenotype. By their nature, rare variants may be in strong association with other non-causative, rare variants across the genome, resulting in the pattern of synthetic association that was observed in the current study (reviewed by [24]). To account for the expected spurious associations left after applying a Bonferroni threshold, an additional test for statistical non-independence among associated SNPs was applied. However, simulation experiments have demonstrated that SNPs causing phenotypic change may not have the lowest *p*-value for association with phenotype in GWAS [24]. Therefore, identifying the true genomic location of these QTL will require follow-up experimentation.

Of the 25 QTL with consensus map locations (14 QTL influencing trait mean and variance across location years and 11 location year-specific), a total of 10 QTL overlapped with the locations of previously reported milling quality QTL. Most prominently, Qkernel.CORE.4D influenced trait means across location years for all traits except Br% and was associated with trait means within most location years (Appendix A). An association between Qkernel.CORE.4D and TWT variance was also observed. The location of this QTL, on Chromosome 4D at 195.7 to 212.1 cM, overlaps with that of QvarPlumps.CORE.4D, which influenced Plumps variance among location years in this study. This genomic location also corresponds to the location of the QTL detected by De Koeyer et al. [10] in the Tera × Marion bi-parental population. MY-A-5 (Milling Yield), PK-A-5 (Plump Kernels), TK-A-5 (Thin Kernels) and KW-A-5 (Kernel Weight) can be placed on the consensus map distal to marker aco118a (at 177.9 cM). Although the QTL interval in the De Koeyer et al. [10] study is substantially broader than in the current study, the remarkably consistent set of associated traits supports the theory of a single underlying QTL with a pleiotropic effect on milling quality. Furthermore, Zimmer et al. [20] reported an association with kernel length (avgbs_200007.1.34 at 211.2 cM) in this genomic region in a set of elite oat lines largely developed by Brazilian breeding programs. The Tera x Marion population was segregating for the *N1* locus controlling the covered/hulless character which mapped to this region [10]. *N1* has been proposed to be a regulatory gene influencing lignan in the developing spikelets [25] and would thus be a plausible candidate gene for differences in milling quality [10]. Although there are very few hulless lines in the CORE, they are included in the homozygous rare allele carriers of Avgbs_cluster_7805.1.9.

The QTL QBr%.CORE.6A influenced the average percent of broken groats over all location years. This SNP was also associated with Br% within all available location years at a *p*-value < 0.01 (Appendix A). Although support for this QTL’s influence on Br% is not available in the literature, several QTL for correlated traits have been reported. A QTL-influencing seedling response to crown rust infection in the CORE association mapping panel was identified on Chromosome 6A at 135.5 (GMI_DS_LB_270) [26]. Crown rust incidence has been positively correlated with the percent of broken groats [5], but this relationship cannot fully explain the influence of QBr%.CORE.6A since crown rust was not observed in the Idaho locations in 2010, 2011 or 2017. Additionally on chromosome 6A, QTL influencing kernel area, kernel length and grain β-glucan content were mapped in the Kanota × Marion population between markers placed at 116.9 to 135.5 cM (cdo836A to cdo665B) which overlaps with QBr%.CORE.6A [17,27]. In the Kanota × Marion population, the Kanota alleles contributed increased kernel area and length but decreased beta-glucan. In the CORE association mapping panel, a QTL influencing β-glucan content (5.1; GMI_ES01_c13021_254) was detected at 135.5 cM [28], based on data from Aberdeen 2010, Fargo 2010 and Lacombe 2010 only. Rare allele homozygotes at avgbs_2_58834.1.20 (QBr%.CORE.6A) had higher percent broken groats compared with the common allele homozygotes (8.66% and 6.55%, respectively). The rare allele was also correlated with slightly lower β-glucan levels in Aberdeen 2010 and Fargo 2010 but not Lacombe 2010. While it should be noted that Br% and β-glucan content were not evaluated in the same location years within the CORE, this pattern is consistent with reports of a negative correlation (−0.63) between Br% and β-glucan [14]. Although it is attractive to assign the Br% and β-glucan effects to the same underlying QTL based on consistent trait correlation, these overlapping QTL may be linked but independent.

Three statistically independent but linked QTL were detected on chromosome 3D that influenced GC variance across location years (QvarGC.CORE.3D.1, QvarGC.CORE.3D.2 and QvarGC.CORE.3D.3). Although based on a fairly large set of location years, the variance estimates used in these association analyses are specific to this data set, and extrapolations beyond this should be made with caution. In addition to GC variance, markers in the QvarGC.CORE.3D.1 QTL influenced TWT in Ottawa 2010, TWT variance and Plumps variance. Qvar.GC.CORE.3D.3 also influenced GC in Saskatoon 2011 (Appendix A). Via et al. [29] summarized on-going theories about phenotypic plasticity and proposed two genetic models: allelic sensitivity wherein the allelic effect on phenotype varies by environment and gene regulation wherein regulatory loci respond to environmental variation by upregulating or downregulating other genes. Variance QTL mapping studies in oat are not available for comparison. However, studies in barley suggest that, as was observed in this study, QTL influencing variance among environments for quantitative traits such as yield and TKW can coincide with the locations of QTL influencing trait means [30,31]. Kraakman et al. [30] theorized that where QTL influencing barley yield stability co-located with QTL influencing yield, then the most likely genetic model would correspond to allelic sensitivity. This genetic model appears to be a particularly good fit with the effects of QvarGC.CORE.3D.3, where a strong effect on GC was observed in Saskatoon 2011 while *p*-values for association in other location years were >0.01.

QTL QvarPlumps.CORE.7D, QvarTWT.CORE.7D and QTWT.Fargo11.7D were identified at overlapping locations (85.2 to 88.7 cM) on chromosome 7D. Although grain width and TWT were not significantly correlated in this study, these results suggest a shared genetic control. QvarPlumps.CORE.7D also influenced TWT in Lacombe 2010 and Saskatoon 2011. Rare allele homozygotes at avgbs2_12115.1.19 (QvarPlumps.CORE.7D) had higher variance across location years (133.04 compared with 88.86 for common allele homozygotes), but Plumps did not differ between the two homozygous classes. Thus, QvarPlumps.CORE.7D appears to fit the genetic model pattern of regulatory gene control of trait stability [29,30]. Additionally mapped to this location was a QTL influencing the heading date in most, but not all, location years in which the CORE was evaluated [21]. In addition, a nearby kernel area and kernel length QTL was mapped in the Kanota × Ogle population at approximately 118.5 cM (cdo1340) [17]. In contrast to the other variance QTL in this region (and those on chromosome 3D), homozygous carriers of the avgbs_cluster_9292.1.49 (QvarTWT.CORE.7D) had lower variance than did the common allele homozygotes (2664.98 and 2791.91, respectively). Such disparate effects in close proximity could complicate efforts to select for more stable milling quality.

Three other mapped QTL detected within a single location year were supported by overlapping location with milling quality QTL reported in the literature. On chromosome 2A, QTKW.Abery10.2A was identified at 34 cM (avgbs_cluster_7015.1.27), which overlaps with the location of a QTL influencing TWT mapped in the Kanota × Ogle population at 32.3 cM (linked to marker umn4090) [16]. The avgbs_cluster_7015.1.27 SNP was also associated with TKW mean across location years but at a *p*-value (*p* = 1.38 × 10^−6^) slightly above the significance threshold. The QPlumps.Saska10.7A QTL (at 61.4 cM) likely overlaps the location of QTL influencing the GC, TKW and TWT in two populations evaluated by Herrmann er al. [7]. Although difficult to place precisely, the region of linkage in Herrmann et al. [7] is probably near 58.5 cM (isu1755A). The Avgbs2_181490.2.45 genotype (QPlumps.Saska10.7A) was also associated with Plumps at *p* < 0.001 in all location years, except Aberdeen 2017, and with mean Plumps across location years. It is worth noting that despite clear evidence for effect sizes that differ between environments, there was no evidence that QTKW.Abery10.2A or QPlumps.Saska10.7A influenced trait variance. Finally, QGC.Lacom11.3C at 35.1 cM is supported by a GC QTL in the Kanota × Ogle population at 33.1 to 32 cM (linked to markers bcd1360A and umn128) [16]. This QTL was also associated at *p* < 0.0001 with GC variance.

When breeding for milling quality, stable expression of phenotype regardless of environment could allow for selection under fewer environments. Increasing the phenotypic stability would also help stakeholders to predict grain characteristics in untested environments. However, Peltonen-Sainto et al. [32] found that the oat lines with greatest yield stability across field locations in Finland were those with the lowest yield. Here, too, GC mean and variance, Plumps mean and variance, Thins mean and variance and TWT mean and variance were all negatively correlated. One key objective in this study was to understand the potential to select lines with high quality that would prove stable across environments. In Rye, the heritability of TKW and TWT variances were estimated at 25 to 30%, respectively, and genomic selection algorithms were able to predict line TKW variance and TWT variance across environment with prediction accuracy of 0.69 and 0.17, respectively [33]. Marker-assisted or genomic selection strategies would require QTL with effect on variance but not mean or a QTL with a favorable effect on both. Twelve QTL influenced the variance of a single trait but were not associated with the mean of the same trait (*p* ≤ 0.01; Figure 1). The rare allele homozygotes at four such QTL (QvarPlumps.CORE.5C, QvarPlumps.Unk2, QvarPlumps.Unk4, QvarPlumps.Unk6) had lower trait variance than the common allele homozygotes, suggesting the potential to select for trait stability without impacting trait mean. The examination of the direction of effect of the rare homozygous class of QvarTWT.CORE.7D, QvarGC.CORE.Unk3, QvarGC.CORE.Unk4, QvarPlump.CORE.Unk1 and QvarOlump.CORE.Unk3, however, suggests that the selection for trait stability using variance QTL would be complicated. Each of these QTL were identified by association with the variance of a single trait at a Bonferroni-adjusted *p*-value threshold, but a post-hoc examination suggested influences on other trait variances at the less conservative *p*-value threshold of 0.01 (Appendix A). The direction of the effect of rare alleles at three of the five QTL were mixed, wherein the variances of some traits were lower while the variances of others were higher in rare allele homozygotes (Figure 1). For example, variances of both Plumps and Thins were higher in the rare allele homozygotes of QvarPlumps.CORE.Unk1, but the TKW variance was lower. The rare allele homozygote class of QvarGC.CORE.Unk4, however, had higher variances of GC, Plumps and TKW, indicating that there may be some regions of the oat genome that could be selected to improve multiple aspects of milling quality stability.

To address the potential to simultaneously select for improved milling quality and stability across environments, QTL were identified which were associated with trait mean or variance at a Bonferroni-adjusted *p*-value threshold and also associated with other trait means and/or variances at the less conservative *p*-value threshold of 0.01 (Appendix A). The direction of the effect of QTL rare alleles on both trait means and variances were examined (Figure 1). In no case would the selection for the rare allele have the potential to improve both quality and stability. The rare allele homozygotes at Qkernel.CORE.4D and QvarTWT.CORE.2D have higher TWT means at the expense of higher TWT variance, while this is true for the common allele homozygotes at QvarTWT.CORE.Unk2. The common allele homozygotes at QvarGC.CORE.3D.1, QvarGC.CORE.3D.3, QvarGC.CORE.Unk1 and QvarGC.CORE.Unk2 had lower GC variance and higher GC means. However, the favorable alleles at these QTL already appear to be quite common in North American elite oat breeding lines (Figure 1).

These milling quality characteristics are complex quantitative traits influenced by numerous QTL with small effects on phenotype so we were interested in the potential to combine favorable QTL. For each trait, we identified the associated QTL (*p* < 0.01). Of those, we considered the additivity of a subset of three or more QTL with >10 rare allele homozygotes and <10% missing data to ensure that at least five lines were homozygous carriers of the allele favorable to the trait at >1 QTL. We were not able to test the additivity of Br%, Plumps, Thins or TKW QTL because too few fit the criteria. To test the additivity of the GC QTL (Qkernel.CORE.4D, QvarGC.CORE.3C and QGC.TWT.CORE.Unk associated with GC at *p* = 1.27 × 10^−36^, 1.22 × 10^−5^ and 1.67 × 10^−13^, respectively), we identified 46 lines homozygous for the allele associated with higher GC at 1 QTL, 308 lines with 2 QTL and 11 lines with 3 QTL. All lines with complete data carried at least one favorable QTL. The number of QTL for which a line was a homozygous rare allele carrier was associated with GC mean (*p* = 4.33 × 10^−38^) and variance (*p* = 2.59 × 10^−4^). GC mean increased and variance decreased with increasing number of favorable QTL carried by a line (Figure 2). To test the additivity of TWT QTL (Qkernel.CORE.4D, QvarTWT.CORE.2D, QvarGC.CORE.3D3, QvarTWT.CORE.Unk2 and QGC.TWT.CORE.Unk associated with TWT at *p* = 6.29 × 10^−29^, 4335 × 10^−4^, 1.36 × 10^−4^, 0.0059, and 1.94 × 10^−8^, respectively), we identified 1 line with 1 QTL, 44 lines with 2 QTL, 209 lines with 3 QTL, 82 lines with 4 QTL and 7 lines with 5 QTL. All lines with complete data carried at least one favorable QTL. The number of QTL for which a line was a homozygous rare allele carrier was associated with the TWT mean (*p* = 0.0030) but not variance (*p* > 0.01). The TWT mean increased with the number of QTL (Figure 3).

## 5. Conclusions

The 57 QTL observed in this study represent a considerable advance in our knowledge of the genetic variation contributing to oat milling quality characteristics. Few studies, to date, have reported QTL influencing these complex quantitative traits. A more complete understanding of the genetic architecture underlying milling quality traits may prove useful to the oat breeding community in their efforts to produce improved cultivars.

## Figures and Tables

**Figure 1 foods-10-02479-f001:**
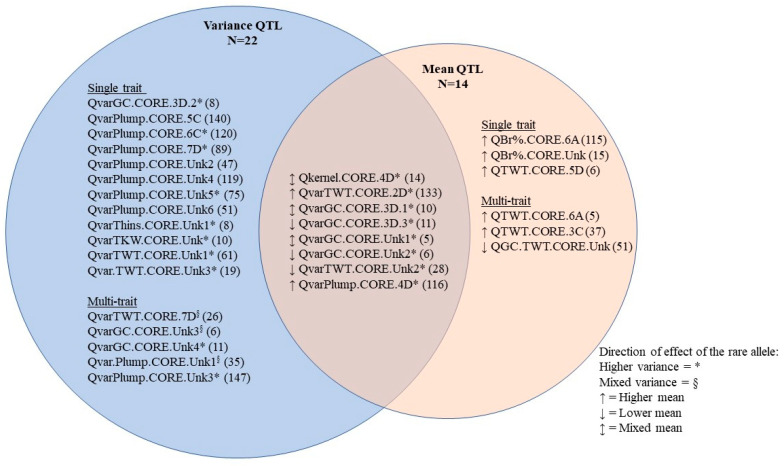
QTL grouped by influence over means and variances of milling quality characteristics. QTL are named using traits for which association at *p* ≤ 1.41 × 10^−6^ and classified as influencing a single trait or multiple traits based on associations at *p* ≤ 0.01. Direction of effect of the rare allele summarizes all associated traits. The number in parentheses is the number of rare allele homozygotes in the sample.

**Figure 2 foods-10-02479-f002:**
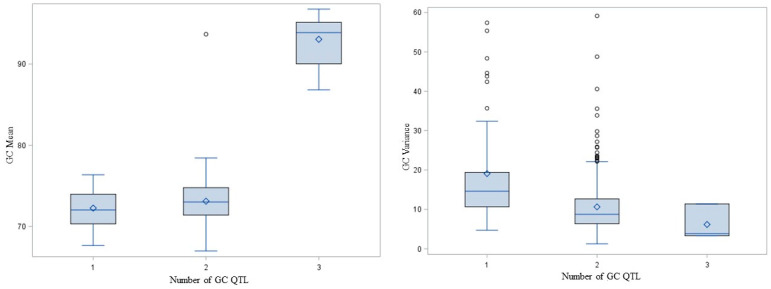
Box and whisker plot showing the mean, quartiles, maximum and minimum GC means and variances of CORE oat lines carrying 1, 2 or 3 copies of QTL influencing GC (Qkernel.CORE.4D, QvarGC.CORE.3D.3, QGC.TWT.CORE.Unk).

**Figure 3 foods-10-02479-f003:**
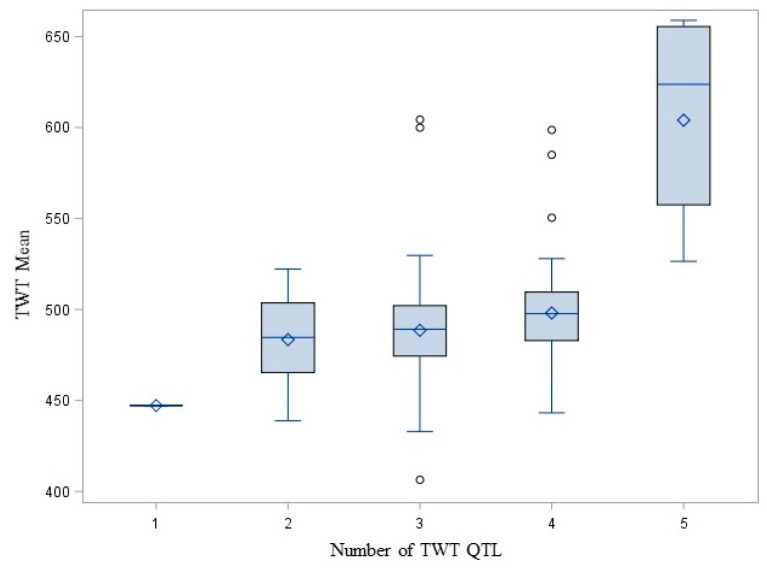
Box and whisker plot showing the mean, quartiles, maximum and minimum TWT means of CORE oat lines carrying 1, 2, 3, 4 or 5 copies of QTL influencing TWT (Qkernel.CORE.4D, QvarTWT.CORE.2D, QvarGC.CORE.3D.3, QvarTWT.CORE.Unk2, QGC.TWT.CORE.Unk).

**Table 1 foods-10-02479-t001:** Mean (Standard deviation) and number of observations (N) of percent broken groats (Br%), groat content (GC), percent plump kernels (Plumps), percent thin kernels (Thins), thousand kernel weight (TKW) and test weight (TWT) for grain from location years used in this study.

	Br%		GC		Plumps		Thins		TKW		TWT	
Location Year		N		N		N		N		N		N
Ad10	NA	NA	NA	NA	NA	NA	22.25 (15.61)	498	NA	NA	533.70 (15.62)	497
Ad17	6.73 (5.19)	470	71.37 (3.42)	468	71.61 (20.27)	487	9.57 (12.10)	487	33.07 (4.30)	493	474 (33.73)	491
Ay10	NA	NA	NA	NA	NA	NA	NA	NA	30.74 (4.49)	499	NA	NA
Fa10	NA	NA	NA	NA	NA	NA	8.25 (10.60)	499	NA	NA	467.37 (32.27)	499
Fa11	4.99 (3.22)	486	73.41 (3.52)	486	NA	NA	14.87 (14.14)	499	NA	NA	447.98 (51.93)	491
It10	NA	NA	NA	NA	NA	NA	NA	NA	NA	NA	466.23 (35.52)	464
La10	NA	NA	NA	NA	NA	NA	18.08 (15.67)	500	37.24 (4.18)	500	570.25 (31.95)	500
La11	2.72 (2.71)	490	75.95 (2.54)	490	NA	NA	NA	NA	NA	NA	NA	NA
Ot10	NA	NA	NA	NA	NA	NA	8.85 (8.45)	398	39.15 (6.15019)	488	501.33 (44.76)	465
Sa10	NA	NA	68.34 (3.48)	483	72.12 (21.48)	496	10.64 (13.23)	496	32.97 (5.03)	496	496.38 (44.84)	496
Sa11	NA	NA	72.62 (4.35)	501	63.32 (24.58)	501	10.46 (14.45)	501	32.35 (4.37)	501	513.03 (31.09)	498
Te10	18.48 (10.48)	387	73.15 (2.90)	387	72.14 (15.52)	395	10.27 (10.66)	395	NA	NA	442.87 (23.42)	394
Te11	4.98 (2.64)	486	75.57 (2.66)	486	61.57 (13.81)	490	15.28 (13.81)	496	NA	NA	451.55 (23.30)	489

**Table 2 foods-10-02479-t002:** Pearson phenotypic correlation coefficients for pairwise comparisons of milling quality trait means and variances (upper cells) with corresponding Prob > |r| (lower cells) for correlations with *p*-values ≤ 0.01.

	Br% Mean	Br% var	GC Mean	GC Var	Plumps Mean	Plumps Var	Thins Mean	Thins Var	TKW Mean	TKW Var	TWT Mean	TWT Var
Br% mean		−0.10	0.07	0.08	0.08	−0.03	−0.03	0.12	0.10	0.12	−0.08	−0.05
Br% var	ns		0.03	0.06	0.11	−0.00	−0.04	0.03	0.07	0.07	−0.09	−0.07
GC mean	ns	ns		−0.36	−0.16	−0.06	0.12	0.04	−0.02	0.12	0.71	0.06
GC var	ns	ns	<0.0001		−0.19	0.11	0.18	−0.10	−0.17	0.07	−0.40	0.08
Plumps mean	ns	ns	0.0004	<0.0001		−0.35	−0.93	0.40	0.76	−0.21	0.01	−0.10
Plumps var	ns	ns	ns	ns	<0.0001		0.33	0.07	−0.24	0.10	−0.04	0.07
Thins mean	ns	ns	0.0078	<0.0001	<0.0001	<0.0001		−0.45	−0.81	0.20	−0.07	0.08
Thins var	ns	ns	ns	ns	<0.0001	ns	<0.0001		0.35	−0.04	0.11	−0.00
TKW mean	ns	ns	ns	0.0002	<0.0001	<0.0001	<0.0001	<0.0001		0.05	0.09	−0.06
TKW var	ns	ns	0.0053	ns	<0.0001	ns	<0.0001	ns	ns		−0.00	0.02
TWT mean	ns	ns	<0.0001	<0.0001	ns	ns	ns	ns	ns	ns		−0.12
TWT var	ns	ns	ns	ns	ns	ns	ns	ns	ns	ns	0.0060	

**Table 3 foods-10-02479-t003:** Summary of mapped quantitative trait loci associated with milling trait means and variances across location years. Consensus linkage map location, trait(s) associated at *p* ≤ 1.41 × 10^−6^ and the name and *p*-value of the representative marker are indicated. *p*-values for the representative marker from analyses of all traits are provided in Appendix A.

QTL	Chr	cM Range ^a^	Traits	Representative Marker ^b^	*p*-Value ^c^
QTWT.CORE.6A	6A	81.7–81.9	TWT	Avgbs_309316.1.34	2.09 × 10^−9^
QBr%.CORE.6A	6A	135.5	Br%	Avgbs2_58834.1.20	1.15 × 10^−7^
QTWT.CORE.3C	3C	70.2	TWT	Avgbs_cluster_24174.1.59	6.06 × 10^−8^
QvarPlumps.CORE.5C	5C	25–28.8	Plumps variance	Avbgs2_146190.1.36	1.13 × 10^−7^
QvarPlumps.CORE.6C	6C	70.5	Plumps variance	Avgbs_6K_109589.1.51	1.73 × 10^−10^
QvarTWT.CORE.2D	2D	151.9	TWT variance	Avgbs_cluster_27091.1.40	3.63 × 10^−12^
QvarGC.CORE.3D.1	3D	45.4–47	GC variance	Avgbs_cluster_22727.1.27	6.47 × 10^−8^
QvarGC.CORE.3D.2	3D	45.4	GC variance	Avgbs_311274.1.63	1.24 × 10^−7^
QvarGC.CORE.3D.3	3D	49.3–49.8	GC, GC variance	Avgbs_cluster_44297.1.21	8.09 × 10^−7^
QvarPlumps.CORE.4D	4D	200.9	Plumps variance	Avgbs_cluster_35424.1.22	6.00 × 10^−12^
Qkernel.CORE.4D	4D	195.7–212.1	GC, Plump, Thins, TKW, TWT, TWT variance	Avbgs_cluster_7805.1.9	1.13 × 10^−51^
QTWT.CORE.5D	5D	26.4	TWT	Avgbs_405598.1.29	3.81 × 10^−7^
QvarPlumps.CORE.7D	7D	85.2	Plumps variance	Avgbs2_12115.1.19	2.85 × 10^−8^
QvarTWT.CORE.7D	7D	85.2–88.7	TWT variance	Avgbs_cluster_9292.1.49	1.77 × 10^−7^

^a^ Range of cM locations of the mapped markers which are statistically non-independent of the representative marker. ^b^ The representative marker is the marker that has placement on the consensus map and the lowest *p*-value for association. ^c^ Lowest *p*-value for association with trait means and variances across location years.

**Table 4 foods-10-02479-t004:** Summary of uncertain genomic location quantitative trait loci associated with milling trait means and variances across location years. The number of non-independently associated SNPs, the number of linkage groups represented by those SNPs, trait(s) and the *p*-value for the representative marker are indicated. *p*-values for all markers associated with these QTL in all experiments are presented in Appendix A.

QTL	# SNPs	# LG ^a^	Trait	Representative Marker ^b^	*p*-Value ^c^
QvarTWT.CORE.Unk1	56	14	TWT variance	Avgbs2_111600.1.12	4.47 × 10^−17^
QvarTWT.CORE.Unk2	66	13	TWT variance	Avgbs2_158635.1.16	2.00 × 10^−16^
QvarPlumps.CORE.Unk1	79	17	Plump variance	Avgbs_16668.1.22	8.02 × 10^−16^
QvarPlumps.CORE.Unk2	9	4	Plump variance	Avgbs_7838.1.46	1.19 × 10^−13^
QGC.TWT.CORE.Unk	1	0	GC, TWT	Avgbs2_92005.1.47	1.67 × 10^−13^, 1.94 × 10^−8^
QvarTWT.CORE.Unk3	4	4	TWT variance	Avgbs2_88377.1.31	8.65 × 10^−13^
QvarGC.CORE.Unk1	54	8	GC variance, TWT	Avgbs2_134662.1.14	7.60 × 10^−8^, 1.60 × 10^−8^
QvarThins.CORE.Unk1	15	5	Thins variance	Avgbs2_147274.1.11	2.54 × 10^−11^
QvarPlumps.CORE.Unk3	2	2	Plumps variance	Avgbs_cluster_20790.1.15	2.76 × 10^−11^
QvarPlumps.CORE.Unk4	13	6	Plumps variance	Avgbs2_31344.1.24	6.37 × 10^−11^
QvarPlumps.CORE.Unk5	19	9	Plumps variance	Avgbs_cluster_18028.1.19	2.11 × 10^−10^
QvarGC.CORE.Unk2	7	0	GC variance	Avgbs2_198262.1.30	2.04 × 10^−9^
QvarTKW.CORE.Unk	2	0	TKW variance	Avgbs2_33384.1.6	7.84 × 10^−9^
QvarGC.CORE.Unk3	12	7	GC variance	Avgbs2_198688.1.12	1.83 × 10^−8^
QvarGC.CORE.Unk4	2	0	GC variance	Avgbs_45323.1.23	2.37 × 10^−8^
QvarPlumps.CORE.Unk6	4	3	Plumps variance	Avgbs2_23356.1.24	2.22 × 10^−7^
QBr%.CORE.Unk	1	0	Br%	Avgbs2_60654.1.6	2.92 × 10^−7^

^a^ The number of linkage groups represented by the set of non-independently associated SNPs. ^b^ The representative marker is the marker with no, or ambiguous, placement on the consensus map and the lowest *p*-value for association with trait means and variances across location years. ^c^ Lowest *p*-value for association with trait means and variances across location years.

**Table 5 foods-10-02479-t005:** Summary of quantitative trait loci associated with milling trait means in single location years. The number of non-independently associated SNPs, the number of linkage groups represented, trait (and context) and the *p*-value for the representative marker are indicated. *p*-values for all markers associated with these QTL in all experiments are presented in Appendix A.

QTL	# SNPs	# LG ^a^	Chr (cM)	Trait (Location Year)	Representative Marker ^b^	*p*-Value ^c^
QTWT.Saska11.Unk1	3	3	NA	TWT (Sa11)	Avgbs_38660.1.16	9.89 × 10^−17^
QTWT.Farge11.Unk1	9	3	NA	TWT (Fa11)	Avgbs_581244	4.48 × 10^−10^
QTWT.Ottaw10.Unk1	1	0	NA	TWT (Ot10)	Avgbs_413385.1.50	7.56 × 10^−11^
QTWT.Fargo11.2D.1	3	1	2D (142.3)	TWT (Fa11)	Avgbs_cluster_39595.1.45	1.63 × 10^−9^
QTWT.Fargo11.7D	1	1	7D (87.3)	TWT (Fa11)	Avgbs_cluster_1682.1.6	2.67 × 10^−9^
QGC.Lacom11.Unk1	241	16	NA	GC (La11)	Avgbs_cluster_17564.1.10	1.05 × 10^−8^
QTWT.Fargo11.Unk2	1	0	NA	TWT (Fa11)	Avgbs_288442	1.83 × 10^−8^
QTWT.Teton11.7A	1	1	7A (36.3)	TWT (Te11)	Avgbs_99656.1.17	2.24 × 10^−8^
QGC.Lacom11.Unk2	1	1	NA	GC (La11)	Avgbs_392008.1.17	3.05 × 10^−8^
QTWT.Saska11.Unk2	1	0	NA	TWT (Sa11)	Avgbs2_104381.1.22	3.33 × 10^−8^
QPlumps.Aberd17.2A	1	1	2A (59.3)	Plumps (Ad17)	Avgbs2_190926.2.35	4.11 × 10^−8^
QGC.Lacom11.3C	5	1	3C (35.1)	GC (La11)	Avgbs2_40053.2.56	4.93 × 10^−8^
QGC.Teton11.Unk	2	2	NA	GC (Te11)	Avgbs2_191851	7.16 × 10^−8^
QTWT.Fargo11.Unk3	1	0	NA	TWT (Fa11)	Avgbs_cluster_15112.1.10	8.56 × 10^−8^
QPlumps.Saska10.7A	1	1	7A (61.4)	Plumps (Sa10)	Avgbs2_181490.2.45	1.16 × 10^−7^
QTWT.Teton10.Unk	15	8	NA	TWT (Te10)	Avgbs2_130697.1.9	1.20 × 10^−7^
QTWT.Ottaw10.Unk2	4	2	NA	TWT (Ot10)	Avgbs_34428	2.60 × 10^−7^
QTKW.Abery10.2A	1	1	2A (34)	TKW (Ay10)	Avgbs_cluster_7015.1.27	2.97 × 10^−7^
QPlumps.Saska11.Unk	3	2	NA	Plumps (Sa11)	Avgbs_4871	3.07 × 10^−7^
QThins.Saska10.Unk	1	0	NA	Thins (Sa10)	Avgbs_cluster_20577.1.16	6.25 × 10^−7^
QBr%.Lacom11.4C	2	1	4C (52.8)	Br% (La11)	Avgbs_cluster_14607.1.44	6.73 × 10^−7^
QTWT.Ithac11.Unk	1	0	NA	TWT (It10)	Avgbs_75885.1.16	6.97 × 10^−7^
QTWT.Saska11.Unk3	1	0	NA	TWT (Sa11)	Avgbs_32105.1.60	7.20 × 10^−7^
QTWT.Teton10.1D	1	1	1D (118.9)	TWT (Te10)	Avgbs2_80976.1.6	7.48 × 10^−7^
QPlumps.Teton10.3A	2	1	3A (104.9)	Plumps (Te10)	Avgbs_cluster_3791.1.28	7.33 × 10^−7^
QTWT.Fargo11.2D.2	1	1	2D (142)	TWT (Fa11)	Avgbs2_115576.1.14	9.08 × 10^−7^

^a^ The number of linkage groups represented by the set of non-independently associated SNPs. ^b^ The representative marker is the marker with no, or ambiguous, placement on the consensus map and the lowest *p*-value for association with trait means and variances across location years. ^c^ Lowest *p*-value for association with trait means and variances across location years.

## Data Availability

Raw data is available at the T3/oat database (http://triticeaetoolbox.org/oat/).

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
