# Peer review of "The Genetic Architecture of Milling Quality in Spring Oat Lines of the Collaborative Oat Research Enterprise"

_foods, 2021, doi:10.3390/foods10102479_

Round 1

Reviewer 1 Report

It is an interesting and important manuscript for me to understand the correlation between phenotype and QTL genotype. The authors try to build the genetic architecture of traits related to milling quality by identifying quantitative trait loci (QTL). However, the label of the samples in this study is hard to follow. For example, I cannot understand the mean of the labeling like QvarPlumps.CORE.7D, QvarTWT.CORE.7D, etc. I am not the major on QTL of oat. I think it should more explain their labels or easily method to make readers understand. For the result and discussion, I do not have negative comments on these sections.

Author Response

I have added an explanation of the naming convention (see lines 212-213).

Reviewer 2 Report

The authors present interesting results on loci associated with important oat quality traits in the CORE panel. The GWAS methods and results are clearly described. However, the phenotypes need further description and analysis per se. For example:

  • Distribution of phenotypes
  • Description of phenotypic models for estimating genotype BLUEs/BLUPs
  • Variance components for location, year, genotype, GxE, error, etc.
  • Heritability of traits
  • Table 2: include correlation coefficients for all pairwise comparisons, not just those that are significant
  • Table 2: are these correlations between plot values (phenotypic correlation) or between genotype BLUEs/BLUPs (genotypic correlation)?

Author Response

The descriptions of phenotypes have been clarified, including:

1) New Supplemental Figure 1 showing the distributions of phenotypes (line 148).

2) Clarification of the use of arithmetic means on line 151.

3) More detailed explanation of effect estimation on line 156 and reporting of genotype r-squares (explained variance), and genotype and location year p-values on lines 196 - 204.

4) Table 2: added correlation coefficients for non-significant pairwise comparisons and clarified that these are phenotypic correlations.